

# Rapid Flood Mapping from Aerial Imagery Using Fine-Tuned SAM and ResNet-Backboned U-Net

Hadi Shokati[1], Kay D. Seufferheld[2], Peter Fiener[2], and Thomas Scholten[1, 3]

[1] University of Tübingen, Department of Geosciences, Soil Science and Geomorphology, Germany
[2] Institute of Geography, Augsburg University, Augsburg, Germany
[3] University of Tübingen, Cluster of Excellence Machine Learning: New Perspectives for Science, Germany

*Correspondence to*: Hadi Shokati (hadi.shokati@uni-tuebingen.de)

**Abstract:** Flooding is a major natural hazard that requires a rapid response to minimize the loss of life and property and to facilitate damage assessment. Aerial imagery, especially images from unmanned aerial vehicles (UAVs) and helicopters, plays a crucial role in identifying areas affected by flooding. Therefore, developing an efficient model for rapid flood mapping is essential. In this study, we present two segmentation approaches for the mapping of flood-affected areas: (1) a fine-tuned Segment Anything Model (SAM), comparing the performance of point prompts versus bounding box (Bbox) prompts, and (2) a U-Net model with ResNet-50 and ResNet-101 as pre-trained backbones. Our results showed that the fine-tuned SAM performed best in segmenting floods with point prompts (Accuracy: 0.96, IoU: 0.90), while Bbox prompts led to a significant drop (Accuracy: 0.82, IoU: 0.67). This is because flood images often cover the image from edge to edge, making Bbox prompts less effective at capturing boundary details. For the U-Net model, the ResNet-50 backbone yielded an accuracy of 0.87 and an IoU of 0.72. Performance improved slightly with the ResNet-101 backbone, achieving an accuracy of 0.88 and an IoU of 0.74. This improvement can be attributed to the deeper architecture of ResNet-101, which allows more complex and detailed features to be extracted, improving U-Net's ability to segment flood-affected areas accurately. The results of this study will help emergency response teams identify flood-affected areas more quickly and accurately. In addition, these models could serve as valuable tools for insurance companies when assessing damage. Moreover, the segmented flood images generated by these models can serve as training data for other machine learning models, creating a pipeline for more advanced flood analysis and prediction systems.

**Keywords:** Flood, ResNet, SAM, UAV, U-Net

## 1 Introduction

Floods are among the world's most common, pervasive, and expensive natural disasters (Smith et al., 2014; Tingsanchali, 2012). Floods cause the greatest number of fatalities, according to the United Nations (Kuenzer et al., 2013). Global warming is expected to increase the frequency and intensity of floods, further amplifying their devastating impacts (Kamilaris and Prenafeta-Boldú, 2018). Recent catastrophic floods in Australia (Kelly et al., 2023), Japan (Lin et al., 2020), Spain (Ezzatvar and López-Gil, 2024) and Germany (Lehmkuhl et al., 2022) underline



that flooding is not limited to the countries of the Global South. However, in countries where the infrastructure for
35   flood protection is inadequate, the loss of human life is significantly higher. In the Global North, economic losses are
generally higher than in the Global South (Taguchi et al., 2022), as long as no major infrastructure projects such as
dams are affected. Effective flood management is therefore essential to mitigate these losses.

Flood management involves four stages: prevention, preparedness, response, and recovery (Plate, 2002). Remote
sensing data plays a critical role in the response and recovery stages. This data is particularly valuable for assessing
40   damage and providing actionable information to speed up recovery efforts, such as guiding insurance claims or
resource allocation (De Leeuw et al., 2014). In recent years, satellite data from platforms such as Sentinel-2, Landsat
8 and Landsat 9 have been widely used for flood-affected area detection (Portalés-Julià et al., 2023). Satellite imagery
provides valuable information to identify flood zones, assess damage and support flood forecasting and risk
assessment models. However, due to revisit limitations, such data is not always available immediately after flood
events. As a dense cloud cover is often associated with flood events, it can be challenging to obtain cloud-free images
after floods. To overcome such limitations, images taken from UAVs and helicopters can provide an effective
alternative to satellite data for flood detection (Hashemi-Beni et al., 2018). UAVs and helicopters can capture high-
resolution images at lower altitudes and are less susceptible to cloud interference. They are also cost-effective and
offer greater flexibility in the scheduling of surveys (Klemas, 2015; Shokati et al., 2023, 2024; Sugiura et al., 2005;
Yao et al., 2019).

Rapid damage assessment is crucial for flood response, as delays can exacerbate the humanitarian and economic toll.
However, recording and processing large volumes of aerial images in real time requires efficient and accurate
automated methods. To counter this, conventional machine learning models such as Support Vector Machine (SVM),
Random Forest (RF) and Maximum Likelihood Classifier (MLC) are often used for flood detection (Tanim et al.,
2022). However, these models face challenges such as reliance on manual feature engineering, inability to capture
complex features such as textural patterns and neglect of spatial correlations in data. To overcome these challenges,
deep learning models have proven to be a powerful tool, demonstrating remarkable effectiveness in various computer
vision applications such as image classification (Jackson et al., 2023; Qiao et al., 2024), object detection (Ye et al.,
2023), and image segmentation (Shokati et al., 2025; Zhang et al., 2023). In particular, numerous studies have
leveraged segmentation techniques to identify areas affected by flooding, demonstrating their potential to address real-
world challenges. For example, (Pally and Samadi, 2022) developed a flood image classifier using different
convolutional neural network (CNN) architectures for segmentation and object detection to calculate water levels and
flood areas. In another study, Safavi and Rahnemoonfar, (2023) compared the performance of different encoder-
decoder and two-pathway architectures to segment flood-affected areas. Similarly, Wieland et al., (2023) investigated
the use of CNNs in detecting water bodies from high-resolution remote sensing imagery. In addition, Wagner et al.,
(2023) compared 32 CNNs for water segmentation using a dataset of 1128 images depicting river water surfaces.

Deep learning-based approaches in hydrology and especially in flood management have made significant progress in
recent years ( e.g. Pan et al., 2019; Wang et al., 2020). However, many models still rely on large datasets for training,
which can affect their performance if the data are limited, unbalanced or specialized (e.g. Mallah et al., 2022). The





model may memorize its features in such cases, leading to overfitting (Safonova et al., 2023). Data augmentation and transfer learning are the most commonly used methods to work with small or highly specialized datasets (Safonova et al., 2023). In data augmentation, new samples are generated by applying different transformations, with the choice of method depending on the type, quality and quantity of data, and transfer learning uses models pre-trained on large datasets, adapting their learned features to new tasks (Safonova et al., 2023). An example of the application of these

techniques to soil erosion is presented by (Shokati et al., 2025) who fine-tuned the SAM, a model pre-trained on a large dataset with over 1 billion masks from 11 million images, to segment erosion and deposition features in agricultural fields. Their approach demonstrated that despite the complexity of erosion and deposition processes and their detection, the fine-tuned SAM model achieved high performance. Another transfer-learning approach uses a residual U-Net architecture (Ronneberger et al., 2015), which improves segmentation performance by utilizing pre-

trained features. U-Net is widely known for its effectiveness in segmentation tasks such as hydrological streamline detection (Xu et al., 2021) and sea-land segmentation (Shamsolmoali et al., 2019). Incorporating residual connections, as implemented by Onojeghuo et al., (2023), can further improve feature propagation and model convergence. The availability of pre-trained deep neural networks such as ResNet (He et al., 2016) trained on large datasets such as ImageNet (Krizhevsky et al., 2012) facilitates cross-domain knowledge transfer, for example, from natural image

classification to remote sensing image segmentation. In particular, combining CNN architectures such as U-Net with ResNet and transfer learning has improved performance on complex tasks such as water body segmentation (Ghaznavi et al., 2024) and plant mapping (Onojeghuo et al., 2023).

Despite significant progress in applying transfer learning in various areas of computer vision, its application to small aerial image datasets for flood management is still largely unexplored. The primary goal in this context is to map the

extent and location of flooded areas. Rapid data collection with helicopters and UAVs — for which only a small dataset is required — makes it possible to use this approach to evaluate preventive and preparatory measures before and during flood events. Furthermore, integrating satellite data could eventually lead to a comprehensive flood forecasting and monitoring system.

The central aim of this study is to test the potential of two advanced transfer learning techniques - the fine-tuned SAM

and the U-Net architecture with ResNet-50 and ResNet-101 backbones - for automated and fast flood area detection and mapping. By automating the mapping process, our approach aims to speed up damage assessment and enable authorities to respond more efficiently and mitigate the financial and human toll of flooding. The priority is to record the extent and location of the flooded areas to assess the extent of the damage and predict possible further damage. Specifically, we address the following research questions:

1.   Which combination of prompt (bounding box or point prompt) for SAM and backbone (ResNet-50 or ResNet-101) for U-Net provides the best performance in flood area detection?

     2.   How do the fine-tuned SAM and U-Net architecture differ in terms of segmentation accuracy for flood area detection using UAV and helicopter imagery?

     3.   How do the elements of the sky, such as clouds or open sky, affect segmentation accuracy?



## 2 Materials and methods

### 2.1 Dataset

This study uses a flood area dataset comprising 290 images with their corresponding masks acquired from Karim et al., (2022). The images were captured using UAVs and helicopters with optical sensors in different regions, including flood events in southern Germany (2013), Karnataka, Kerala and Maharashtra in India (2019), Sabah in Malaysia (2021) and Bangladesh (2022). The dataset includes a variety of scenes, including rural areas, urban areas, peri-urban areas, greenery, buildings, mountains, rivers, sky, and roads, with masks created using Label Studio software (Figure 1).

Due to the use of different platforms and sensors, the imaging conditions were inherently inconsistent. The camera angle during imaging was inconsistent. In some cases, no clouds or sky elements were visible due to the low camera angle, while in other cases, sky elements such as clouds or open sky were visible due to greater camera rotation. In addition, the images were taken at different heights, resulting in different spatial resolutions. The image dimensions also varied considerably: the width ranged from 219 to 3648 pixels and the height from 330 to 5472 pixels. To ensure the uniformity of the dataset for modeling purposes, all images were resized to 256 × 256 pixels prior to analysis.

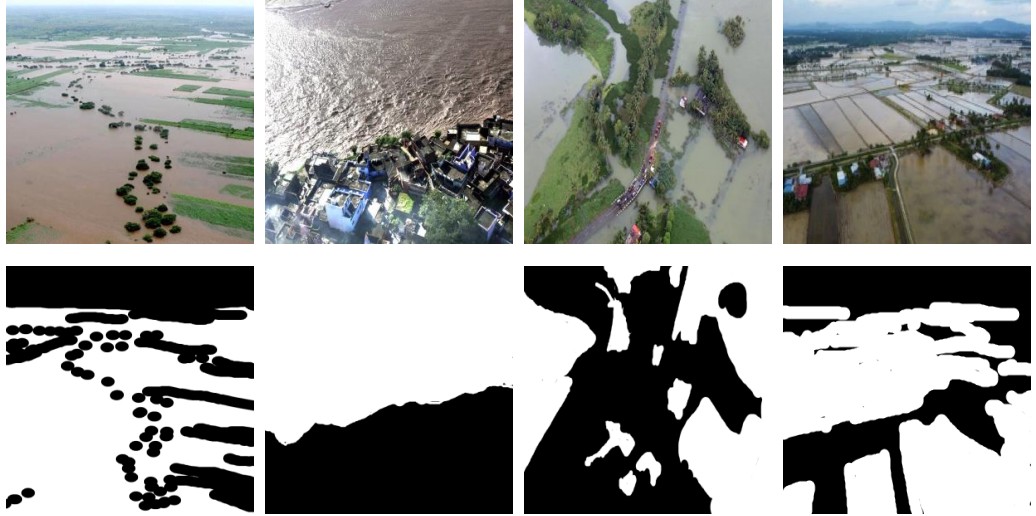

**Figure 1: Example images from the Flood Area dataset (top) with their corresponding ground truth masks (bottom). Images and their corresponding masks are from Karim et al., (2022).**

### 2.2  Network Architecture

#### 2.2.1 Fine-tuning Segment Anything Model

The Segment Anything Model (SAM) developed by Meta AI Inc., USA is an image segmentation model trained on 1 billion masks extracted from 11 million images. As the name suggests, it can segment any image without the need for



additional training data (Kirillov et al., 2023). The architecture of SAM is composed of three primary components: an image encoder, which is built on a robust Vision Transformer (ViT) backbone and extracts features from the input image; a prompt encoder, which uses the input prompts to create embeddings; and a mask decoder, which generates the final mask by combining the outputs of the previous components (Figure 2).

The SAM architecture enables the integration of human prompts, which increases the efficiency of annotation by the human in the loop. The prompts guide the model to focus on specific regions of the image, improving segmentation accuracy. These prompts can take different forms, e.g., bounding boxes (Bbox), points and texts. To implement prompt-based interaction with the SAM architecture, we used both Bbox and point prompts in a fully automated manner. For the Bbox prompts, we first identified all foreground pixels in the annotated images (pixels representing flood). The Bbox was then computed as the smallest rectangle enclosing all foreground pixels in each binary mask by determining their minimum and maximum x and y coordinates. For the point prompts, we randomly selected 30 foreground pixels per image. To ensure spatial diversity and avoid clustering, a minimum Euclidean distance of 10 pixels was set between two selected points. This restriction contributed to a more representative coverage of the flooded area, which in turn improves segmentation accuracy.

SAM offers several variants, each tailored to different computational requirements and based on distinct configurations of the Vision Transformer (ViT) backbone: ViT-Base, ViT-Large, and ViT-Huge, containing approximately 91M, 308M, and 636M parameters, respectively (Kirillov et al., 2023). Upon evaluating these variants, we observed that their effectiveness in detecting flood-affected areas was remarkably comparable. To optimize our computational resources, we opted for the ViT-Base variant as it offers a favorable balance between performance and efficiency.

Although SAM can process any image without additional training, its performance may be limited in complex tasks. In such cases, fine-tuning can enhance its segmentation accuracy (Shokati et al., 2025). Fine-tuning is a transfer learning technique that applies a pre-trained model, such as SAM, which has already learned general patterns from a large dataset. A task-specific dataset, such as a flood dataset, is then prepared and formatted to match the model's input requirements. Certain layers of the model are either frozen or modified to regulate the extent to which training updates alter the original knowledge. The model is then trained with the new dataset at a lower learning rate to refine its understanding without overwriting prior knowledge. After training, the model's performance is evaluated using a validation set, and hyperparameters are adjusted for optimization. Once the model achieves satisfactory accuracy, it is deployed and continuously monitored to ensure robust performance on real-world data. In this study, the mask decoder of SAM (Figure 2b) is fine-tuned (modified) as it has a simple and efficient design that requires fewer computational resources (Sun et al., 2024). This ensures that the fine-tuning process is fast, efficient, and requires less memory. In this process, the other two components, the image encoder (Figure 2a) and the prompt encoder (Figure 2c) are kept fixed (frozen), meaning their parameters are not updated during fine-tuning. This ensures that only the mask decoder is modified.



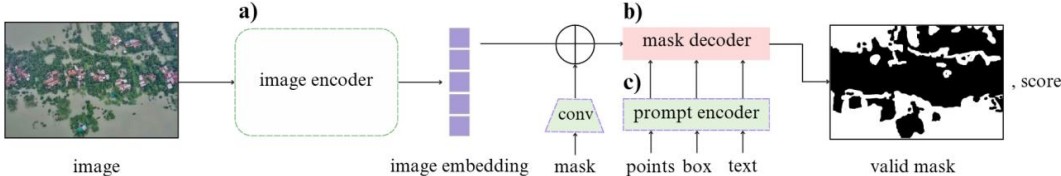

**Figure 2: Schematic figure of the Segment Anything Model architecture, where the image encoder processes the image to**
**extract features, which are combined with the prompt encoder's embeddings and passed to the mask decoder to generate**
**the final segmentation mask. Image and mask are from Karim et al., (2022).**

### 2.2.2 U-Net architecture with ResNet as the backbone

U-Net is a fully convolutional neural network introduced in 2015 for segmenting biomedical images (Ronneberger et
al., 2015). Over time, it has been adapted for other fields, such as remote sensing. With its U-shaped architecture, U-
Net comprises a down-sampling path (encoder) and an up-sampling path (decoder). This structure enables the
convolutional network to learn and combine features at various levels of detail, which is critical for accurately
segmenting small regions and fine details in images.

The down-sampling path follows a typical convolutional network architecture. It involves repeated applications of
3×3 convolutions (without padding), each followed by a ReLU (Rectified Linear Unit) activation and a 2×2 max-
pooling operation with a stride of 2 to reduce the spatial dimensions. At each down-sampling stage, the number of
feature channels doubles, enabling the network to extract increasingly complex features while progressively losing
spatial information. This part of the network is crucial for capturing high-level features from the input image.

The up-sampling path is the reverse of the down-sampling path. This part of the network utilizes deconvolution
operations to double the spatial dimensions of the feature maps. Following each deconvolution, a concatenation
operation is performed with the corresponding feature maps from the down-sampling path to restore spatial
information. This process helps recover the details lost during the down-sampling phase and enhances the network's
spatial accuracy for final segmentation. Skip connections are used to concatenate feature maps from the corresponding
layers in the encoding path to the decoding layers, ensuring the recovery of information lost during down-sampling.
At the end of the architecture, a 1×1 convolutional layer is applied to map the feature maps to the desired number of
segmentation classes. This final layer assigns a class label to each pixel, producing the segmented output.

To achieve better results with limited data, we apply knowledge from transfer learning in this study. Specifically, we
use a residual neural network (ResNet) (He et al., 2016) pre-trained on ImageNet as the encoder backbone of U-Net.
The weights of the pre-trained encoder are kept frozen to utilize the existing low-level feature representations (such
as edges, corners, and textures), while only the U-Net decoder is trained on our dataset. This allows the decoder to
adapt to our segmentation task without updating the encoder's weights (Figure 3).

ResNet is a convolutional neural network (CNN) that uses identity skip connections to solve the degradation issue that
occurs when accuracy reaches saturation and rapidly deteriorates as network depth grows. Convolutions of $1 \times 1$, $3 \times$



3, and 1 × 1 series are used to stack several bottleneck residual blocks to create it. There are several variations of the
ResNet model depending on the network's depth, including ResNet18, ResNet34, ResNet50, ResNet101, and others.
ResNet50 and ResNet101, which have 50 and 101 layers, respectively, were used in this study.

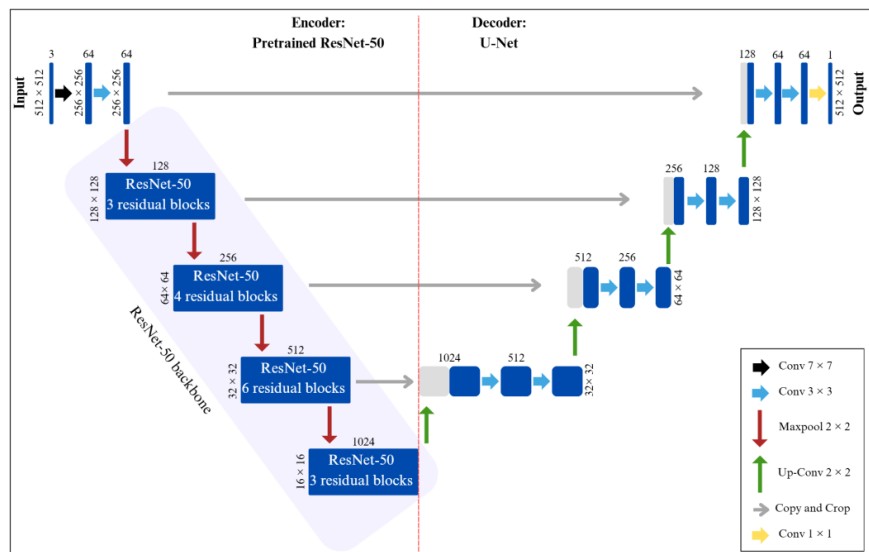

**Figure 3: Architecture of U-Net with ResNet-50 backbone (adapted from Manos et al., (2022))**

### 2.3 Experimental Training Setup

For the segmentation model, 70% of the flood area dataset (204 samples) was used for training. To enhance the
diversity of the training data, we applied several data augmentation techniques. These included geometric
transformations, such as random horizontal and vertical flips and rotations, as well as color-based transformations like
random grayscale conversion and Gaussian blur. The model was trained for 50 epochs, with each epoch representing
a complete pass the training data through the model. A batch size of 4 and a learning rate of $1e^{-3}$ were used. A learning
rate scheduler was employed to adjust the learning rate if the validation loss did not decrease over 10 consecutive
epochs, reducing the previous learning rate by multiplying it by a factor of 0.1. Additionally, 15% of the dataset (43
samples) was used for model validation at the end of each epoch. Among the 50 models generated, the one with the
lowest validation loss was selected as the best model and tested on the remaining dataset (15%, 43 samples). The
Adam optimizer algorithm was utilized due to its strong adaptability (Liu et al., 2023). To minimize the divergence
between the predicted and the observed values, we used DiceCELoss, a loss function that integrates Dice Loss with
Cross-Entropy Loss (CE Loss). DiceCELoss is often used to improve segmentation performance by leveraging both
the pixel-wise accuracy (via Cross-Entropy) and the structural similarity (via Dice coefficient).

The dice and cross-entropy losses are calculated as:

$$L_{Dice} = 1 - \frac{2\sum_{c=1}^{C}\sum_{i=1}^{N} g_i^c s_i^c}{\sum_{c=1}^{C}\sum_{i=1}^{N} g_i^c + \sum_{c=1}^{C}\sum_{i=1}^{N} s_i^c} \tag{1}$$



$\qquad L_{CE} = -\frac{1}{N}\sum_{c=1}^{C}\sum_{i=1}^{N} g_i^c \ \log S_i^c$ (2)

Where $L_{Dice}$ and $L_{CE}$ are dice and cross-entropy losses, respectively, N is the number of pixels, $g_i^c$ is the ground truth binary indicator of the class label c of pixel i, and $S_i^c$ is the corresponding predicted segmentation probability.

Experiments were carried out using Python and the PyTorch framework on a Windows 11 computer equipped with an NVIDIA GeForce RTX 4070 Ti GPU.


**2.4 Model performance evaluation**

The performance of segmentation models is usually evaluated using several metrics, e.g. intersection-over-union (IoU), dice coefficient, recall, precision and accuracy. IoU quantifies the overlap between the predicted and true flood-affected regions, with higher values indicating better model performance in terms of accurately identifying flood-

affected areas. The dice coefficient is similar to IoU, but emphasizes correct positive predictions, making it particularly useful in scenarios with class imbalance. Recall measures the model's ability to correctly identify all flood-affected regions, with higher values reflecting fewer overlooked regions. Precision evaluates the accuracy of the model's positive predictions and indicates the proportion of correctly identified flood-affected regions out of all predicted positive outcomes. Finally, accuracy provides an overall measure of the correctness of the model, considering both

correct predictions of flood-affected and non-flood-affected regions. However, it may not be as informative for highly imbalanced datasets. Each of these metrics provides different insights into the performance of the model and can be used together for a comprehensive assessment. They are given in the following equations (Vinayahalingam et al., 2023):

$\text{IoU} = \frac{TP}{TP+FP+FN}$ (3)

$\text{Dice coefficient} = \frac{2\,TP}{2\,TP+FP+FN}$ (4)

$\text{Recall} = \frac{TP}{TP+FN}$ (5)

$\text{Precision} = \frac{TP}{TP+FP}$ (6)

$\text{Accuracy} = \frac{TP+TN}{TP+TN+FP+FN}$ (7)

Where TP, TN, FP, and FN stand for the pixel labels for true positives, false positives, true negatives, and false

negatives, respectively.



# 3 Results and discussion

## 3.1 Training and validation losses

Both U-Net (with ResNet-50 and ResNet-101 as backbones) and fine-tuned SAM (with Bbox and point prompts) models were trained for 50 epochs (Figure 4). For the SAM models, whether trained with points or Bbox prompts, the training loss exhibited a sharp decline at the beginning of the training process. This was followed by a slower, more gradual decrease until the final epochs. Similarly, the validation loss initially experienced a steep decline, which then slowed progressively until the point where the downward trend almost stopped, especially for Bbox prompts where the validation loss even increased very slightly, which could be a sign of overfitting (Shorten and Khoshgoftaar, 2019). A comparison of training and validation losses between the points prompt and the Bbox prompt revealed that the model trained with the points prompt generally performed better in both learning the training set and generalizing to the validation set.

For the U-Net model, both training loss and validation loss gradually decreased throughout the training process, whether ResNet-50 or ResNet-101 was used as the backbone. This indicates that the model effectively learned the relationships within the training set while also improving its ability to generalize to unseen data (validation set). The training and validation loss values were nearly identical when using ResNet-50 and ResNet-101 as backbones, suggesting that both backbones performed similarly in terms of learning and generalization capabilities.

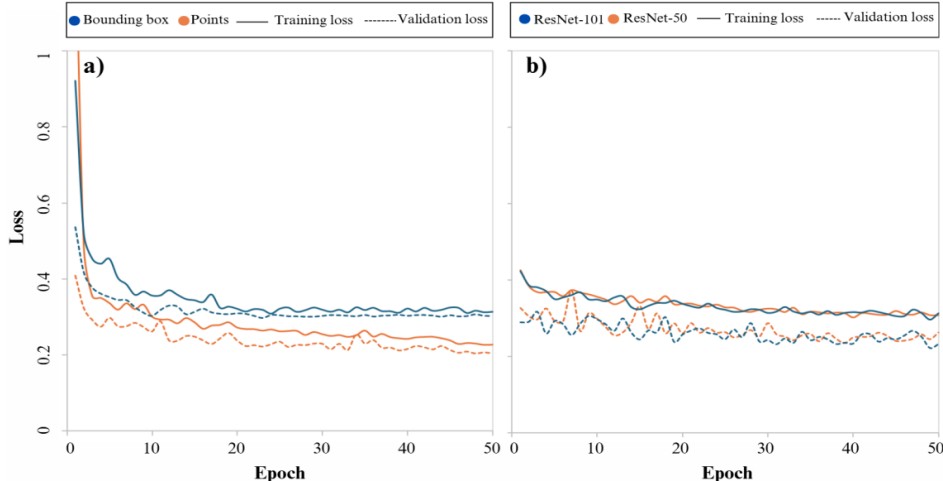

**Figure 4: Training and validation losses for a) Segment Anything Model with points and bounding box prompts and b) U-Net model with ResNet-50 and ResNet-101 backbones**

## 3.2 Performance of the model on the validation set

For the SAM model with points prompt, all evaluation metrics consistently increased during the initial epochs, followed by stabilization in the later epochs (Figure 5a). The SAM model with Bbox prompts demonstrated a similar




trend, with metrics steadily increasing before reaching stability (Figure 5b). Comparing these two prompting methods revealed that the SAM model with points prompt (Figure 5a) outperformed the version with Bbox prompts (Figure 5b) regarding overall validation performance.

For the U-Net model, the metrics for the ResNet-50 backbone showed a steady improvement, eventually reaching stabilization (Figure 5c). The U-Net model with the ResNet-101 backbone followed a similar trajectory, with only minor differences in terms of stability (Figure 5d). A direct comparison of the two backbone configurations revealed that both ResNet-50 and ResNet-101 backbones performed similarly in learning and generalization for the U-Net model.

When comparing the performance of the SAM model to the U-Net model, regardless of the prompt type or backbone configuration, the SAM model consistently outperformed the U-Net model on the validation set.

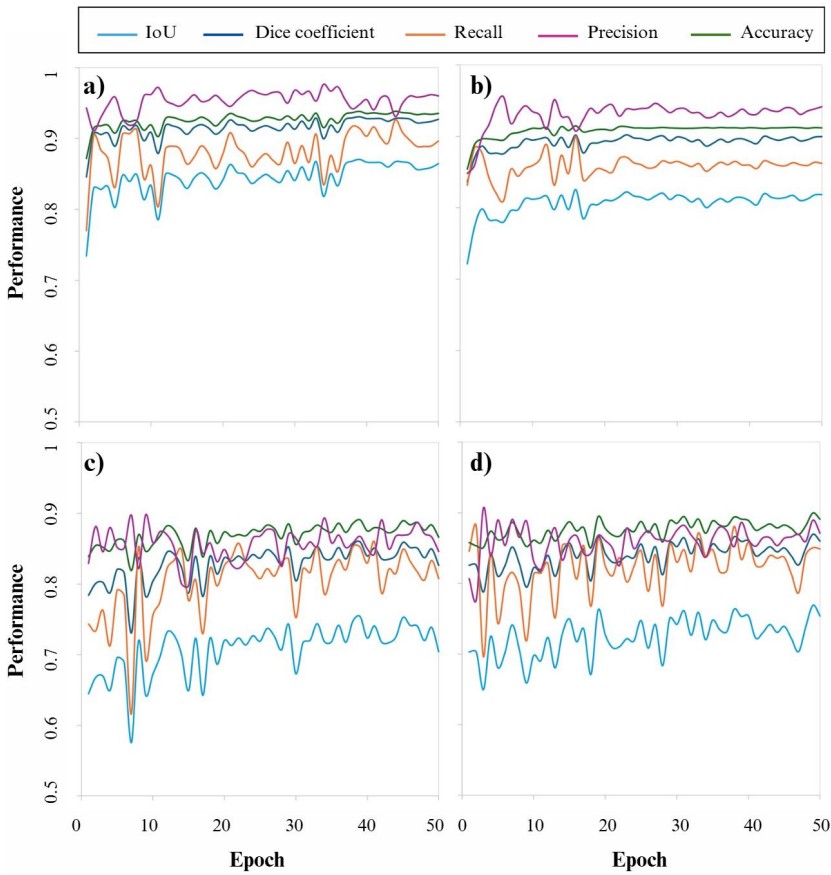

**Figure 5: Performance of the Segment Anything Model on validation set using a) points prompt and b) bounding box prompt and the performance of U-Net model with c) ResNet-50 and d) ResNet-101 backbones**






### 3.3 Evaluating segmentation results

In this study, the performance of the SAM and U-Net models was evaluated for segmenting flooded areas. The SAM model was assessed with two types of prompts, points prompt and Bbox prompt, while the U-Net model was tested with two backbone types, ResNet-50 and ResNet-101. The results of the different metrics are shown in Table 1, and

examples of the segmentation results of all models are shown in Figure 6.

The results indicated that the SAM model with the points prompt (SAM-Points) outperformed all other models (Table1). The strong performance of this model can be primarily attributed to the ability of the points prompt to provide precise spatial information. This information allows the model to recognize the exact boundaries of the flood-affected areas and to make accurate predictions. This ability is particularly important in regions where light intensity

varies significantly (e.g. Figures 6e and 6h), or the boundaries of flooding are unclear. In addition, the architecture of the SAM model uses input prompts, which increases the accuracy of its predictions and contributes to its superior performance.

The SAM model with the Bbox prompt (SAM-Bbox) showed the weakest performance among all other models (Table 1). Although SAM-Bbox achieved the highest recall (0.96), it performed worse on other metrics compared to SAM-

Points. The higher recall indicates the model's strong ability to identify flood-affected areas. However, the lower precision (0.69) shows that it is difficult to delineate the boundaries accurately, resulting in extraneous or non-contiguous areas being part of the flood-affected areas. This limitation arises from the limited information provided by the Bbox prompt, which only provides the general framework of the target area without offering detailed boundary data. In other words, the Bbox prompts provide the model with limited information about the exact boundaries, which

increases the probability of false-positive predictions.

While the existing literature suggests that Bbox prompts often perform better than point-based prompts in different contexts (Cheng et al., 2023; Gaus et al., 2024; Mazurowski et al., 2023; Xie et al., 2024), the choice of prompt is highly dependent on the specific nature of the dataset. Flood segmentation is a rare case where point-based prompts outperform Bbox prompts. In the analyzed dataset, the flooding areas often extend across the entire image from one

edge to another. Consequently, a Bbox usually covers most of the image and provides the model with less detailed information than point-based prompts. Theoretically, this lower granularity of information from Bbox prompts leads to poorer performance in such cases. However, this observation must be interpreted in light of the specific assumptions and methodological choices made in our study. In particular, both types of prompts were generated fully automatically and without manual refinement, and the point prompts were spatially constrained to ensure dispersion across the

flooded regions. These design decisions, in combination with the inherent characteristics of flood imagery, such as large, amorphous regions that often cover significant portions of the image, have a direct impact on the relative effectiveness of each prompt type. Therefore, the superior performance of point-based prompts observed in our experiments should not be generalized to other domains without carefully considering the characteristics of the dataset properties and the prompt generation strategies.



The U-Net models used in this study showed moderate performance in flood segmentation (Table 1). Notably, the U-Net model with a ResNet-101 backbone outperformed the model with a ResNet-50 backbone. This improved performance of the ResNet-101 backbone was also reported in a study by Ait El Asri et al., (2023) which focused on aerial image segmentation. The superior performance of the ResNet-101 can be attributed to its ability to extract more complex and detailed features from the images due to its deeper layers, allowing U-Net to identify flood-affected areas

with higher accuracy. However, the difference between the two models was minimal, and U-Net with ResNet-101 only outperformed ResNet-50 on a few images (e.g. Figure 6g). The small difference in results suggests that increasing the network depth does not significantly improve performance in the context of flood segmentation. This could be because flooding patterns are not complex enough to require deeper architecture for effective analysis. Nevertheless, the performance of U-Net with both backbones lagged behind that of SAM-Points, highlighting the superiority of the

SAM model in utilizing prompt-based information to improve prediction accuracy. The better performance of the fine-tuned SAM model compared to U-Net was also highlighted in a similar study by Lehmiani et al., (2023) that focused on medical image segmentation.

In summary, regarding the first research question, which investigates the optimal combination of prompts for SAM and backbones for U-Net, the results indicated that the SAM model with the points prompt performed better than the

SAM model with the bounding box prompt. In addition, the U-Net model with the ResNet-101 backbone performed better than the model with ResNet-50. In response to the second research question on the differences in segmentation accuracy between the fine-tuned SAM model and the U-Net model with ResNet-50 and ResNet-101 backbones, our results showed that the SAM-Points model significantly outperformed both U-Net configurations, while the SAM-Bbox model had the weakest performance among all models tested. This analysis shows how the design of the model,

the choice of prompt and the choice of backbone can directly influence segmentation performance.

To answer the research question regarding the effects of sky elements on segmentation accuracy, our analysis revealed that segmentation accuracy of all models decreased in images where sky elements such as clouds or open sky were present due to significant camera rotation angles, compared to conditions where the camera provided a near top-down view (e.g. Figures 6e and 6h). This finding is consistent with the results of (Simantiris and Panagiotakis, 2024). This

is because segmentation methods are often based on color information. If the color of objects such as the sky or clouds is similar to the color of the areas affected by the flood, the model may have difficulty distinguishing them from the flood and incorrectly segment them as part of the flood zone. This problem was less pronounced in the SAM model with the points prompt than in the other models, while it was more prominent in the SAM model with the Bbox prompt, as each Bbox usually covers almost the entire image. In other words, sky elements can be falsely considered as

potential regions for segmentation.



**Table 1: Segmentation results of the test dataset using the Segment Anything model with points and bounding box prompts and the U-Net model with ResNet-50 and ResNet-101 backbones (the highest accuracies are in bold)**

| Model | Accuracy | IoU | Dice | Precision | Recall |
|---|---|---|---|---|---|
| **SAM-Points** | **0.96** | **0.90** | **0.95** | **0.94** | 0.95 |
| SAM-Bbox | 0.82 | 0.67 | 0.81 | 0.69 | **0.96** |
| U-Net (ResNet-50) | 0.87 | 0.72 | 0.84 | 0.83 | 0.85 |
| U-Net (ResNet-101) | 0.88 | 0.74 | 0.85 | 0.83 | 0.87 |



385

390







395



**Figure 6: Example segmented images using the Segment Anything model with points and bounding box prompts and the U-Net model with ResNet-50 and ResNet-101 backbones. Images and Ground truths are from Karim et al., (2022).**



### 3.4 Effectiveness of the models in real-world applications

The results of this study highlight the effectiveness of U-Net (with ResNet-50 and ResNet-101 backbones) and fine-tuned SAM (with Bbox and point prompts) models in segmenting flood-affected areas. Despite variations in flight altitude, image quality, resolution and viewing angles, these models performed remarkably well. In real-world scenarios, where such variations are common, these models can be expected to process aerial images of affected regions quickly and without additional training. This rapid prediction capability can help emergency teams immediately identify flooded areas and take the necessary measures. In addition, these models are of great benefit to insurance companies in assessing damage, speeding up insurance payments and improving post-crisis services. In summary, these approaches offer significant benefits for flood crisis management and help minimize human and economic flood impacts through timely and precise interventions.

### 4 Conclusion

In this study, we used two advanced transfer learning techniques: the fine-tuned SAM model and the U-Net architecture with ResNet-50 and ResNet-101 backbones to detect flood-affected areas. Using the Flood Area dataset, which includes UAV and helicopter aerial images of flood-affected areas, we aimed to evaluate and compare the performance of these models in accurately segmenting flood-affected areas. Our results showed that the fine-tuned SAM model outperformed the U-Net model when point prompts were used, while it performed worst among all models when Bbox prompts were used. This emphasizes the crucial role of prompting strategies in influencing the performance of the fine-tuned SAM model in flood segmentation. The results of the study offer practical benefits for emergency response teams as they allow for a faster and more accurate assessment of the areas affected by the floods. In addition, the models are also of great value to insurance companies in the damage assessment phase. Despite these promising results, there is still room for further research. Future work could include the development of a user-friendly interface that allows emergency responders and insurance professionals to seamlessly utilize these models and effectively interpret the results. In addition, extending the models to predict high-risk areas before flooding occurs - using inputs such as topographical data, rainfall trends and river flow information - could further enhance their utility.

**Author contribution**

**HS:** Conceptualization, Data curation, investigation, Methodology, Formal analysis, Software, Visualization, Writing – original draft. **KDS:** Writing – review and editing. **PF:** Funding acquisition, project administration, Data curation, Conceptualization, Supervision, Writing – review and editing. **TS:** Funding acquisition, Supervision, Conceptualization, project administration, Data curation, Writing – review and editing.

**Competing interests**

The authors declare that they have no known competing financial interests or personal relationships that could have appeared to influence the work reported in this paper.



**Financial support**

This work was supported by the Deutsche Forschungsgesellschaft (DFG) [No. SCHO 739/25-1].

**Code and data availability**

The codes and data are available in the GitHub repository.

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
