# Peer review of "Rapid Flood Mapping from Aerial Imagery Using Fine-Tuned SAM and ResNet-Backboned U-Net"

_EGUsphere, 2025_

## Author Comment (AC1)

**Revision Notes, egusphere-2025-3146**

Dear Editor and Reviewers,

We would like to express our sincere gratitude for your time and thoughtful comments on our manuscript, " **Rapid Flood Mapping from Aerial Imagery Using Fine-Tuned SAM and ResNet-Backboned U-Net**." Your insightful feedback has been extremely valuable in helping us improve the clarity, strength, and overall quality of our work.

We have carefully considered all suggestions and addressed them point-by-point in the revised manuscript. For your reference, we have highlighted our responses to your comments in green. We believe these revisions have significantly strengthened the manuscript and we are confident that it is now ready for further consideration.

Thank you again for your valuable contribution to this process. We look forward to your feedback on the revised manuscript.

**The author's reply to the comments is highlighted in green.**

| Comments | Responses |
|---|---|
| The research is well designed and written. It contributes to the development of a strong and user-friendly AI tool that can provide quick and effective support in flood-affected areas where urgent assistance is needed, without requiring harmonized or standardized procedures for image collection from different sources. As a limitation of the research, I believe it would be valuable to suggest including the geolocation of the final flood map to facilitate relief efforts. | Thank you for your careful reading and constructive suggestion. We fully agree that adding geolocation to the final flood maps would substantially increase the usefulness of the system for emergency responders and for insurance loss assessment. We would like to clarify that the dataset used in this study consisted of 290 aerial images and their corresponding manually created masks provided by a third party; these images did not include GPS/INS metadata or any georeferenceable files (e.g., GeoTIFF, orthophotos). Because the original dataset lacks precise location information, it was not possible to produce geolocated outputs in this work. We have now explicitly stated this limitation in the revised manuscript and added a short "future work" plan that describes practical approaches (e.g., collecting GNSS/RTK-enabled UAV imagery, using ground control points and photogrammetric orthorectification, or aligning masks to georeferenced basemaps) to enable georeferenced flood maps in follow-up studies. We appreciate the suggestion and will prioritize geolocation in our future data |

| | collection and system development so that the model outputs can be directly used for field operations and addressing location-specific help requests (Please see lines 422-425). |
|---|---|
| Furthermore, the reasons behind the superiority of SAM-Points should be discussed. Compared to other methods, this approach appears to be more effective in distinguishing bare soil from flooded areas. | Thank you for raising this important point. We agree that further clarification is necessary. In our study, the superior performance of SAM with point prompts over bounding box prompts can be explained by several dataset-specific characteristics. First, in flood imagery, water often extends across the entire scene with highly irregular and amorphous boundaries. Bounding boxes in such cases tend to cover almost the whole image and thus provide little discriminative information to the model, sometimes even introducing ambiguity between flooded and non-flooded regions. By contrast, multiple dispersed point prompts explicitly highlight localized regions within the flood extent and along its boundary, which allows SAM to capture fine-grained differences more effectively (as previously mentioned in the manuscript). Second, flood boundaries are less sharply defined compared to other object segmentation tasks, and point prompts serve as stronger anchors for delineating these diffuse regions. Together with our automatic prompt generation strategy (which ensured dispersed placement of points within flooded areas), these factors explain why SAM-Points outperformed SAM-Bbox in this context. We have revised the manuscript to emphasize these aspects more clearly (Please see lines 307-309). |

---

## Author Comment (AC2)

**Revision Notes, egusphere-2025-3146**

Dear Editor and Reviewers,

We would like to express our sincere gratitude for your time and thoughtful comments on our manuscript, "**Rapid Flood Mapping from Aerial Imagery Using Fine-Tuned SAM and ResNet-Backboned U-Net**." Your insightful feedback has been extremely valuable in helping us improve the clarity, strength, and overall quality of our work.

We have carefully considered all suggestions and addressed them point-by-point in the revised manuscript. For your reference, we have highlighted our responses to your comments in green. We believe these revisions have significantly strengthened the manuscript and we are confident that it is now ready for further consideration.

Thank you again for your valuable contribution to this process. We look forward to your feedback on the revised manuscript.

**The author's reply to the comments is highlighted in green.**

| Comments | Responses | Manuscript Change |
|---|---|---|
| The research is well designed and written. It contributes to the development of a strong and user-friendly AI tool that can provide quick and effective support in flood-affected areas where urgent assistance is needed, without requiring harmonized or standardized procedures for image collection from different sources. As a limitation of the research, I believe it would be valuable to suggest including the geolocation of the final flood map to facilitate relief efforts. | Thank you for your careful reading and constructive suggestion. We fully agree that adding geolocation to the final flood maps would substantially increase the usefulness of the system for emergency responders and for insurance loss assessment. We would like to clarify that the dataset used in this study consisted of 290 aerial images and their corresponding manually created masks provided by a third party; these images did not include GPS/INS metadata or any georeferenceable files (e.g., GeoTIFF, orthophotos). Because the original dataset lacks precise location information, it was not possible to produce geolocated outputs in this work. We have now explicitly stated this limitation in the revised | Despite these promising results, there is still room for further research. **A limitation of the present study is that the Flood Area dataset we used (290 aerial images and associated masks) did not include GPS/georeferenced metadata, so it was not possible to produce delocalized results. Addressing this dataset limitation in future works would enable more accurate and actionable relief.** |

| | | |
|---|---|---|
| | manuscript and added a short "future work" plan that describes practical approaches (e.g., collecting GNSS/RTK-enabled UAV imagery, using ground control points and photogrammetric orthorectification, or aligning masks to georeferenced basemaps) to enable georeferenced flood maps in follow-up studies. We appreciate the suggestion and will prioritize geolocation in our future data collection and system development so that the model outputs can be directly used for field operations and addressing location-specific help requests (Please see lines 423-426). | |
| Furthermore, the reasons behind the superiority of SAM-Points should be discussed. Compared to other methods, this approach appears to be more effective in distinguishing bare soil from flooded areas. | Thank you for raising this important point. We agree that further clarification is necessary. In our study, the superior performance of SAM with point prompts over bounding box prompts can be explained by several dataset-specific characteristics. First, in flood imagery, water often extends across the entire scene with highly irregular and amorphous boundaries. Bounding boxes in such cases tend to cover almost the whole image and thus provide little discriminative information to the model, sometimes even introducing ambiguity between flooded and non-flooded regions. By contrast, multiple dispersed point prompts explicitly highlight localized regions within the | Theoretically, this lower granularity of information from Bbox prompts leads to poorer performance in such cases. **In addition, the inherently diffuse and irregular nature of flood boundaries makes point prompts stronger cues for guiding the model, while bounding boxes typically include both flooded and non-flooded regions, providing the model with less discriminatory guidance.** |

| | flood extent and along its boundary, which allows SAM to capture fine-grained differences more effectively (as previously mentioned in the manuscript). Second, flood boundaries are less sharply defined compared to other object segmentation tasks, and point prompts serve as stronger anchors for delineating these diffuse regions. Together with our automatic prompt generation strategy (which ensured dispersed placement of points within flooded areas), these factors explain why SAM-Points outperformed SAM-Bbox in this context. We have revised the manuscript to emphasize these aspects more clearly (Please see lines 307-310). | |
|---|---|---|
| Upon re-reading the manuscript, I noticed that in lines 200–203 you mention the use of various data augmentation techniques. Could you please clarify the probability settings assigned to each augmentation method? | Thank you for your comment. We have revised the manuscript to clarify the probability settings of the data augmentation methods. The revised text (Lines 201–204) now specifies that random horizontal and vertical flips, rotations (±30°), Gaussian blur, and random grayscale conversion were each applied with a probability of 0.5. | These included geometric transformations such as random horizontal and vertical flips and rotations of **up to 30°** as well as color-based transformations such as random grayscale transformations and Gaussian blurs **with a kernel size of 3, all applied with a probability of 0.5.** |
| In lines 201–203, it is not clear whether the augmentation was applied exclusively to the training dataset. Providing this clarification would enhance the transparency of the methodology. | Thank you for your insightful comment. We confirm that data augmentation was applied exclusively to the training dataset to increase its diversity. This clarification has been added to the revised manuscript (Please see lines 200-201). | Data augmentation was applied **exclusively to the training set** to increase the diversity of the training data. |

| | | |
|---|---|---|
| Still in lines 201–203, it would be highly valuable to explicitly include details regarding the number of images before and after data augmentation, as well as their distribution across the training, validation, and test sets. Such information is critical to ensure reproducibility. | We sincerely thank the reviewer for this valuable comment. We applied data augmentation exclusively to increase data diversity rather than the number of samples. Consequently, the total number of images in each split (training: 204, validation: 43, testing: 43) remained unchanged. This clarification has been incorporated into the revised manuscript (Please see lines 200-201). | Data augmentation was applied exclusively to the training set to increase **the diversity** of the training data. |
| In lines 209–219, you mention the use of both Dice Loss and Cross-Entropy Loss. Could you please specify how these two loss functions were combined? For example, were they summed, averaged, or weighted differently? | We thank the reviewer for pointing this out. The Dice Loss and Cross-Entropy Loss were combined by taking their average. This clarification has been added to the revised manuscript (Please see lines 210-213). | To minimize the divergence between the predicted and the observed values, we used DiceCELoss, a loss function that integrates Dice Loss with Cross-Entropy Loss (CE Loss). **Specifically, the two components were combined by taking their average,** leveraging both the pixel-wise accuracy (via Cross-Entropy) and the structural similarity (via Dice coefficient) to improve segmentation performance. |
| I appreciate that the code is publicly available on GitHub. However, I could not locate the corresponding datasets in the repository. Based on the README file, it seems that the authors expect users to obtain the data from an external source. While this is acceptable provided that the source remains reliably available, hosting a copy of the datasets within your GitHub repository would be | We thank the reviewer for this valuable suggestion. We have now added a direct link to the datasets in our GitHub repository to improve accessibility and ensure long-term availability. The README file has been updated accordingly. | |

| preferable for long-term accessibility. | | |
|---|---|---|

---

## Author Comment (AC3)

Dear Armin,

We would like to express our sincere gratitude for your time and feedback on our manuscript, " **Rapid Flood Mapping from Aerial Imagery Using Fine-Tuned SAM and ResNet-Backboned U-Net**."

**The author's reply to the comment:**

| Comment | Response |
|---|---|
| Dear authors, I read the paper, and I see the method used by the authors, and also the code is derived from the following paper and GitHub: ArminMoghimi/Fine-tune-the-Segment-Anything-Model-SAM-: A. Moghimi, M. Welzel, T. Celik, and T. Schlurmann, "A Comparative Performance Analysis of Popular Deep Learning Models and Segment Anything Model (SAM) for River Water Segmentation in Close-Range Remote Sensing Imagery," https://github.com/ArminMoghimi/Fine-tune-the-Segment-Anything-Model-SAM- https://doi.org/10.1109/ACCESS.2024.3385425 However, I couldn't see this reference in the reference list, which it should be. | We appreciate your feedback. After a thorough review, we did not find any similarities between our code and the one you mentioned. Our implementation originates from our previous article published in *CATENA*, where we focused on segmenting erosion and deposition. That codebase is the result of years of experience and numerous meetings with domain experts. We therefore respectfully clarify that our implementation is fully independent and not derived from the repository you mentioned. We highly recommend reading the article linked below for a better understanding of the foundations of our approach. https://www.sciencedirect.com/science/article/pii/S0341816225002565 We were not aware of your article and code before, but we appreciate your work — it is a valuable contribution to the field. If we had seen it earlier, we would have certainly benefited from it. In the academic community, the shared goal is to advance the state of the art, and we are no exception. To support this goal, we will include a comparison with your results in the results section of our revised manuscript, which |

| | we are confident will further enhance the quality of the paper. |
| --- | --- |
| | Thank you again for your valuable feedback. |

---

## Author Comment (AC5)

**Revision Notes, egusphere-2025-3146**

Dear Armin,

We would like to express our sincere gratitude for your time and feedback on our manuscript, " **Rapid Flood Mapping from Aerial Imagery Using Fine-Tuned SAM and ResNet-Backboned U-Net**."

**The author's reply to the comment:**

| Comment | Response |
|---|---|
| Dear Hadi, Thank you—please don't get me wrong—you are one of our remote sensing community, and your work is good. I don't want to dwell on the similarity aspect, as we've already explored comparisons between SAM and UNet50 (with ResNet backbone) in the context of water segmentation using close-range remote sensing images (from UAVs, smartphones, and handheld cameras, within a 1–300 meter range) (https://doi.org/10.1109/ACCESS.2024.3385425). However, I'd like to suggest that you consider referencing our results, as they align with your findings and could strengthen your discussion section. For instance, we observed that SAM performs very well in general. However, when segmenting images from the same area, UNet actually produced even better results. This nuance might enrich your discussion, especially when highlighting the practical performance differences between models. | Dear Armin, Thank you once again for your valuable feedback. We truly appreciate the time and effort you have taken to engage with our work. In our work, we not only compared U-NET and SAM, but also evaluated two types of input prompts in SAM (points and bounding boxes) and two types of backbones for U-NET (ResNet-50 and ResNet-100). We agree that including a direct comparison with your findings will make our paper more comprehensive, and we will add this comparison to the results section of our revised manuscript. Regarding the ViT backbone, we also evaluated multiple variants and observed that their effectiveness for flood-affected area segmentation was comparable. To optimize computational resources, we selected ViT-Base, as it provides a favorable balance between accuracy and efficiency. |

Another point is related to the computational aspects: SAM typically operates on 1024×1024 patches, and when fine-tuning with a frozen ViT backbone, it still requires significant computational resources. The choice of ViT backbone also matters—ViT-H is quite heavy and not ideal for fine-tuning, whereas the smaller variants (like Tiny ViT and Medium ViT) tend to perform better with fewer resources.

Lastly, regarding datasets: One could argue that with SAM, we might not need large annotated datasets anymore. While SAM reduces the need for manual annotation, I would still say that datasets are necessary. The real question is: how many do we actually need? To help address this, you might consider referencing this recent paper by Professor Anette Eltner from Dresden University: https://doi.org/10.1080/01431161.2025.2457131

It discusses the sensitivity of model performance to dataset size and could help you frame this as a potential advantage of SAM over UNet.

I'd love to hear your thoughts and see how you might incorporate some of these ideas into your discussion.

Warm regards,

Armin

We also appreciate your recommendation of Professor Anette Eltner's recent article. It provides an excellent perspective on dataset size requirements. The number of labeled images needed depends on the complexity of the task. For instance, in our previous study on erosion and deposition segmentation (https://doi.org/10.1016/j.catena.2025.108954) we worked with about 400 labeled images and observed clear performance gains with increasing dataset size, an effect that was particularly relevant given the higher complexity of that task compared to flood mapping. This increased complexity was because eroded and non-eroded soil often have very similar visual characteristics, whereas flooded areas are usually more distinct from their surroundings.

Thank you again for your thoughtful suggestions. They will certainly help us improve the clarity and impact of our paper.

---

## Author Comment (AC6)

**Revision Notes, egusphere-2025-3146**

Dear Editor and Reviewers,

We would like to express our sincere gratitude for your time and thoughtful comments on our manuscript, "**Rapid Flood Mapping from Aerial Imagery Using Fine-Tuned SAM and ResNet-Backboned U-Net**." Your insightful feedback has been extremely valuable in helping us improve the clarity, strength, and overall quality of our work.

We have carefully considered all suggestions and addressed them point-by-point in the revised manuscript. For your reference, we have highlighted our responses to your comments in blue. We believe these revisions have significantly strengthened the manuscript and we are confident that it is now ready for further consideration.

Thank you again for your valuable contribution to this process. We look forward to your feedback on the revised manuscript.

**The author's reply to the comments is highlighted in blue.**

| Comments | Responses | Manuscript Change |
|---|---|---|
| In their paper, Hadi Shokati et al. Propose methods to improve rapid flood mapping from Aerial Imagery using Fine-Tuned SAM and ResNet-Backboned U-Net. This paper is a valuable contribution to remote sensing and rapid disaster assessment. Although none of the comments and suggestions are critical, I would like to ask authors to incorporate and address these issues and suggestions before the paper's publication.

Although the methods in this paper are related to floods, they do not directly discuss flood itself. Therefore, it would be beneficial for the readers and also enhance the paper's visibility to replace "flood" in keywords with "flood mapping" or "segmentation of flood", which are more relevant to the presented study. | We appreciate the reviewer's insightful comment. We replaced 'Flood' with 'Flood Mapping' in Keywords to increase the visibility of the paper. Please see line 27. | **Keywords:** Flood Mapping, ResNet, SAM, UAV, U-Net |
| The introduction and methods sections are well written, | Thank you for your valuable comment. The suggested | To minimize the discrepancy between observed and predicted |

| | | |
|---|---|---|
| addressing the main issues and research question. However, there was a minor absence of the reference in line 210 regarding the choice of DiceCELoss. It is claimed in the paper that: *"DiceCELoss is often used to improve segmentation performance by leveraging both the pixel-wise accuracy (via Cross-Entropy) and the structural similarity (via Dice coefficient)."* Please include at least one reference to support this claim and the choice of this loss function. | reference has been added to the manuscript to support the statement regarding the choice of DiceCELoss. Please see lines 210 - 215. | flood extents, we used the Dice-Cross-Entropy Loss, which averages the Dice loss and cross-entropy loss. This composite loss function is widely used in model training, as it balances the strengths of both components **(Hadlich et al., 2023; Shokati et al., 2025)**. It facilitates rapid convergence and often improves final performance, particularly enhancing the Dice coefficient **(Hadlich et al., 2023),** which is critical for accurately capturing the spatial overlap between predicted and actual flood areas. |
| Although the terms and names of the methods are well described throughout the paper, their usage in the text and figures is inconsistent. For example, Segment Anything Mode is consistently abbreviated as SAM, but the versions (point prompts or points prompts) are referred to in various inconsistent forms. Please use consistent terminology for methods in the entire manuscript, especially for the main methods. For instance, here are a few examples:
• **SAM (Points prompts)** on page 14 and **SAM (Point prompts)** on page 15 in the figures.
• **Points** in Figure 4.
• **point prompts** in line 309
• **point prompt** in line 100
• **"Bounding boxes"** is abbreviated as **"Bbox"** in line 135, but this is inconsistently used throughout the text and figures, sometimes as | We appreciate the reviewer's valuable comment regarding this inconsistency.

In response, we have carefully revised the entire paper to ensure consistency in terminology. Specifically, we unified all variations of the method names:

For Bounding box, we consistently used the full form "**bounding box prompts**" throughout the text.

For Point prompts, we used the full form "**point prompts**" consistently.

For Segment Anything Model, we consistently used its abbreviation "SAM" in the text, however, in the figures, we kept the full form (e.g., "Segment Anything Model (SAM)") to ensure that readers can interpret the figures independently without referring back to the text.

All the revised and standardized terms have been highlighted in blue throughout the manuscript. | All the revised and standardized terms have been highlighted in blue throughout the manuscript. |

| | | |
|---|---|---|
| **"bounding box"** and other times as **"Bbox."**
 • ... | | |
| Figure 6 lacks a brief description of the subplots labeled a, b, ..., h in the caption. | Thank you for your insightful comment. Subplots (a–h) correspond to different samples from the dataset we used. However, to make the caption clearer, we added a brief description. Please see lines 396 - 398. | Figure 6: Example segmented images using the Segment Anything Model with point and bounding box prompts (SAM-Points and SAM-Bbox models, respectively) and the U-Net model with ResNet-50 and ResNet-101 backbones. **Subplots (a–h) correspond to different samples from the dataset of Karim et al., (2022).** |
| I would like to ask the authors to elaborate on why 290 images with different geographic regions and diverse flood events are sufficient for this study. We recognize that transfer learning enables us to train our models with a limited sample size by leveraging pre-trained data; I would appreciate a discussion on how this sample size captures the variability needed for a robust model. Including this clarification would strengthen the manuscript by addressing potential concerns about the dataset. | We thank the reviewer for this comment. In response, we have added a new section to the manuscript.

 In this section, we explain that in transfer learning, the number of labeled samples required depends on task complexity, model architecture, and the similarity between the pre-trained model and the target task. Our dataset of 290 images, covering flood events in Germany, India, Malaysia, and Bangladesh, provides broad geographic and environmental variability. The inclusion of UAV and helicopter imagery with different angles and altitudes, combined with data augmentation techniques, further increases the effective diversity.

 Empirical results show that the fine-tuned SAM model achieved an IoU of 0.90 and an accuracy of 0.96 on unseen images, confirming that the dataset captures sufficient variability for reliable flood segmentation. Comparable studies (e.g., Ghaznavi et al., 2024; Shokati et al., 2025) also demonstrate strong | **3.5 Dataset Size and Diversity Considerations**

 Determining the optimal dataset size in transfer learning does not depend on a fixed number but rather on several factors, including task complexity, model architecture, and the similarity between the pre-trained source domain and the target task. In transfer learning, large-scale pre-trained models such as SAM (Kirillov et al., 2023) and ResNet (He et al., 2016) already capture rich, generalized feature representations from millions of natural images. As a result, a relatively small number of labeled samples is often sufficient for fine-tuning to achieve high performance in specialized applications. Our dataset consists of 290 images covering flood events in countries such as Germany, India, Malaysia, and Bangladesh. This geographic diversity ensures variability in environmental conditions, land cover types, flood characteristics, and illumination. The inclusion of both UAV and helicopter imagery from different camera angles and altitudes further increases this variability, providing a robust basis for model generalization. Additionally, data augmentation techniques (such as rotations, flips, grayscale |

| performance with similar dataset sizes. Please see lines 409 - 425. | transformations, and Gaussian blur) increased the effective training diversity and reduced the risk of overfitting. |
| | Empirically, our results (Table 1) demonstrate that the fine-tuned SAM model achieved an IoU of 0.90 and an accuracy of 0.96 on unseen data, confirming that the dataset sufficiently captured the variability required for reliable flood segmentation. Comparable studies on environmental and remote sensing tasks (e.g., Ghaznavi et al., 2024; Shokati et al., 2025) have reported strong performance using datasets of similar size, reinforcing the suitability of our sample in the context of transfer learning–based flood segmentation. |